# Segmental Rib Index and Spinal Deformity: Scoliogenic Implications

**DOI:** 10.3390/healthcare11223004

**Published:** 2023-11-20

**Authors:** Theodoros B. Grivas, Nikola Jevtic, Danka Ljubojevic, Samra Pjanic, Filip Golic, Elias Vasiliadis

**Affiliations:** 1Department of Orthopedics & Traumatology, “Tzaneio” General Hospital of Piraeus, 185 36 Piraeus, Greece; 2Scolio Centar, 403916 Novi Sad, Serbia; njevticns@gmail.com (N.J.);; 3Department of Pediatric Rehabilitation, Institute for Physical and Rehabilitation Medicine “Dr Miroslav Zotovic“, 78000 Banja Luka, Bosnia and Herzegovina; samra.pjanic@hotmail.com (S.P.);; 43rd Department of Orthopaedics, School of Medicine, KAT Hospital, National and Kapodistrian University of Athens, 165 41 Athens, Greece; eliasvasiliadis@yahoo.gr

**Keywords:** idiopathic scoliosis, double rib contour sign, rib index, rib cage, Cobb angle, segmental rib index, rib hump deformity, Adams test, lateral standing radiographs, angle of trunk rotation, scoliometer

## Abstract

The aim of this report is to evaluate the segmental rib index (RI) from the T1 to T12 spinal levels in mild and moderate idiopathic scoliosis (IS) curves of thoracic, thoracolumbar and lumbar type by gender. The relationship of segmental RI to the frontal plane radiological deformity presented as the Cobb angle and to the posterior truncal surface deformity presented as the scoliometric readings of Angle of Trunk Rotation (ATR) in these patients is also assessed. Any statistically significant relationship between these parameters would be very important for biomechanical relations in rib cage (RC) deformity presented as rib hump deformity (RHD) and deformity in the spine, and would thus provide valuable information about scoliogeny. The segmental rib index (RI) is presented in 83 boys and girls with mild and moderate IS. The measurements include the scoliometric readings for truncal asymmetry (TA), the Cobb angle assessment and the segmental RI from T1-T12. The statistical package SPSS 23 was used for statistical analysis. The TA was documented and the Cobb angle is presented by gender and curve type. The segmental RI of thoracic, thoracolumbar and lumbar curves are presented for the first time. The correlations of the segmental RI to surface deformity presented as rib hump deformity (RHD) in all IS patients, and particularly in thoracic curves, to Cobb angle by gender and age and the comparison of the segmental RI index of asymmetric but not scoliotic children to the scoliotic peers by curve (in thoracic, thoracolumbar, lumbar curves) in boys and girls are presented. The findings emphasize the significant protagonistic role of thoracic asymmetry in relation to the spinal deformity, mainly in girls for the thoracic and in boys for the thoracolumbar curves. The cut-off point of age of the examined scoliotics was 14 years, which is when the RI shows a stronger correlation with spinal deformity, namely when thoracic deformity is decisively effective in the development of thoracic spinal deformity, in terms of Cobb angle. In summary, the results of this study may provide scoliogenic implications for IS, as far as the role of the thorax is concerned.

## 1. Introduction

The existence of truncal asymmetry (TA) presented as rib hump deformity (RHD) in the Adams bending test during the school scoliosis screening (SSS) programs is a finding which defines the number of children who will be referred to the scoliosis outpatient departments of hospitals and is a dominant predictor of scoliosis [1,2,3].

The rib index (RI) method was introduced as a radiological measurement for the assessment of the RHD, which essentially represents rib cage (RC) deformity in the transverse plane. It was originally presented to assess the RHD of IS at the more prominent point of the double rib contour sign (DRCS) on the lateral standing radiographs (LSR) of their RC [4].

The RI is used for the documentation of initial rib-cage deformity on the transverse plane before any treatment, for the assessment of the benefits of physiotherapeutic scoliosis-specific exercises on RC deformity [5], for the assessment of bracing on RC deformity [6] and for the pre- and post-operative assessment of rib-cage deformity and its correction [7,8].

Studying the lateral spinal profile (LSR) of the IS patients radiographically, we recognized that the thoracic level of the most prominent point of the DRCS differs in the various types of IS curves. This observation motivated us to study the RI segmentally at all the vertebral levels (T1–T12) on the DRC in the radiographs of thoracic, thoracolumbar and lumbar curves. We also investigated whether there was any correlation between segmental RI thoracic levels and the degree of the Cobb angle, respectively, regarding when and how the existing surface deformity correlates to mild and moderate spinal deformity.

In idiopathic scoliosis (IS), the word etymology strictly means the factor(s) causing the AIS, pathogenesis means the mode of origin of the morbid process and pathomechanism/pathobiomechanics means the sequence of events in the evolution of its structural and functional changes that result from the pathological process [9]. Prof. RG Burwell of Nottingham suggested the word scoliogeny as the collective noun to include aetiology, pathogenesis and pathomechanism/pathobiomechanics [10]. However, when scoliogeny is used in this study, we will only refer to the pathogenesis and the pathobiomechanics of this condition.

In summary, the aim of this report was to evaluate the segmental RI from the T1 to T12 spinal levels in mild and moderate IS thoracic, thoracolumbar and lumbar curves by gender and to assess the relationship of segmental RI to the frontal plane radiological deformity and to the scoliometer readings of posterior truncal surface deformity presented by the Angle of Trunk Rotation (ATR). Any significant relationship between these parameters would be very important for the biomechanical relations of the RC (RHD) with spine deformity and may thus provide information about scoliogeny.

## 2. Material and Methods

*Study design.* This is a retrospective statistical study on cross-sectional data collected in Greece, Bosnia and Herzegovina and Serbia.

The examined subjects: Eighty-three children and adolescents, twenty boys and sixty-three girls, with juvenile and adolescent IS were included in the study, with a mean age 12.3 ± 2.7 (range 7 to 17 years) and a mean Cobb angle of 23.8 (15.5–38.7) degrees. Of these, 25 patients (19 girls, 6 boys) had primary thoracic, 33 (27 girls, 6 boys) thoracolumbar and 25 (17 girls, 8 boys) lumbar curves, respectively. In addition, 27 asymmetric children who were referred for further radiological assessment due to the presence of RHD at the Adams test with a radiological curve less than 10 degrees or with a straight spine were also included. The data of scoliotics were collected before any treatment. Any non-idiopathic scoliotic case was excluded from the study.

*The measurements.* The Cobb angle was assessed according to the classical method described by Cobb, 1948 [11]. Scoliosis is considered if the Cobb angle is ≥10°, according to SRS. The segmental rib index was measured as follows. Initially, for the determination of the DRC and the calculation of the RI, we determined the LSRs: (a) At the most extending rib contour (convex) for the most extended rib point, which is the contact point of a vertical line tangential to this most extended point, and we drew this vertical line passing from this point. (b) At the least extending rib contour (concave), we determined the most projected rib point, which is the contact point of a vertical line tangential to this most extended point, and we drew a vertical line passing from this point. We determined the posterior margin line of the body of the corresponding vertebra of these two previously noted points, and we measured the distances from the posterior margin line of the corresponding vertebra. These two distances were labeled d1 and d2. The quotient of d1/d2 is the rib index for the vertebral level of the corresponding vertebra; see Figure 1 [4]. The RΙ is likewise assessed in all the above and below vertebral levels of the initial RI measurement. Thus, the segmental RIs from T1 to T12 are calculated; see Figure 2.

Truncal asymmetry (TA) assessment. The RHD was measured using the scoliometer. We previously reported the reliability study for the rib index method [4] and for the scoliometer readings [12].

The measurements for the segmental rib index were made by one of the co-authors, (DL), while the scoliometric measurements were undertaken by TBG, SP, and NJ.

*Statistical analysis*. The statistical analysis was performed using the SPSS 23.0 (SPSS Inc., Chicago, IL). The results were presented as frequency (percent), mean and standard deviation (SD), median and interquartile range (IQR). A Kruskal–Wallis test and t-test were used to compare the groups. A Mann–Whitney test was used as a post hoc test, if there were significant differences between groups. Bonferroni correction was used to adjust the *p*-value for multiple comparisons between sub-groups. Pearson’s correlation coefficient was used to measure the strength of the relationship between the Cobb angle and rib index in the sample and the groups by curve type, by sex and by age. All *p* values less than 0.05 were considered significant.

## 3. Results

The mean age and the Risser stage in the 83 included patients in the study scoliotics are presented in Table 1.

The Cobb angle by gender and curve type of the 83 scoliotic patients in the study is presented in Table 1 and Table 2. It is evident that the IS was mild or moderate, and the sample was suitable to draw conclusions regarding the developing deformity.

The basic characteristics for the 83 studied subjects, in total and by gender, are shown in Table 1. The average age is similar in both groups by gender. The Risser stage was significantly higher in the female group. A slightly higher Cobb angle and ATR were found in the female group, but without statistical significance.

The Cobb angle by gender and curve type of the 83 scoliotic patients of the study is presented in Table 2.

Truncal asymmetry (TA): the mean scoliometer reading was 7.7 ± 4.5 degrees (range: 2 to 13).

### 3.1. Segmental Rib Index and Surface Deformity (RHD) Correlations

In all 83 cases with IS, the measured Cobb angle was significantly correlated to the RI at the T6, T7 and T8 levels; Table 3.

In the 25 thoracic IS curves in both genders the Pearson Correlation coefficient was significant for the T6–T12 levels, but it was not significant at any level of the thoracolumbar and lumbar curves (Table 4).

### 3.2. Segmental Rib Index and Cobb Angle Correlations

The segmental rib index by curve type and gender among all the three curve types—thoracic, thoracolumbar and lumbar—in both male and female patients is shown in Table 5 and Table 6. No significant correlation was found among the three groups.

The same analysis in female patients showed significant differences only between groups at T8, T9 and T12 vertebral levels. The post hoc analysis showed significant differences between thoracic and thoracolumbar curves at T9 and between thoracolumbar and lumbar curves at T12 (Table 5). The same analysis in male patients showed no significant difference between groups (Table 6).

In order to investigate further the effect of growth on the relationship of RC with spinal deformity in terms of the Cobb angle, we analyzed the segmentally measured RI from T1 to T12. For this purpose, the studied scoliotics were split into two age cohorts, namely (a) scoliotics less than 13 and over 13 years of age, and (b) scoliotics less than 14 and over 14 years of age. It was thus possible to recognize accurately the cut-off age of the examined scoliotics at which the segmental RI starts correlating or shows stronger correlation to the spinal deformity, specifically when the thoracic deformity starts to dictate the development of spinal deformity, in terms of Cobb angle.

The comparison analysis of the segmental RI for female patients only in all types of curves in the four age groups showed the following results.

In both cohorts of these age groups together, using the Kruskal–Wallis test, a significant difference was observed only in the older girls over 13 years of age at T10 level (*p* = 0.007), and in the group over 14 years old (*p* = 0.006).

Comparing the segmental RI index to the Cobb angle by the type of curve at all ages for boys and girls, the Pearson Correlation coefficient was significant at T10–T12 for thoracic, T4 for thoracolumbar and T9 for the lumbar curves; see Table 7.

Splitting the ages of the scoliotics in two groups, namely younger and older, and using the Pearson correlation coefficient to compare the RI index to the Cobb angle by curve type in younger and older boys and girls, the following was found.

Comparing the segmental RI index to the Cobb angle by type of curve for boys and girls less than 13 years of age, the Pearson correlation coefficient was significant at T4 for thoracolumbar and T1–T2 for the lumbar curves.

Comparing the segmental RI index to the Cobb angle by type of curve for boys and girls over 13 years of age, the Pearson correlation coefficient was significant at T11–T12 for thoracic, T6-T9 for thoracolumbar and T8, T9 and T11 for the lumbar curves.

Comparing the segmental RI index to the Cobb angle by type of curve for boys and girls less than 14 years of age, the Pearson correlation coefficient was significant at T4 for thoracolumbar and T8 for the lumbar curves.

Comparing the segmental RI index to the Cobb angle by type of curve for boys and girls over 14 years of age, the Pearson correlation coefficient was significant at T8–T11 for thoracic and T1 for thoracolumbar curves (see Table 7).

Comparing the segmental RI index of the asymmetric but not scoliotic children to the scoliotic peers by curve type for boys and girls, interestingly, no significant difference between groups (non-scoliotic to thoracic, non-scoliotic to thoracolumbar, non-scoliotic to lumbar) was found.

## 4. Discussion

To the best of our knowledge (reviewing the literature), the segmental RI is presented for the first time in a sample of thoracic, thoracolumbar and lumbar mild and moderate IS patients, in Table 4 and Table 5.

No full agreement exists on the definition of mild and moderate idiopathic scoliosis. Mild idiopathic scoliosis is characterized in different reports by a Cobb angle of more than 10 and less than 30 degrees [13], of more than 10 to 25 degrees [14], and of more than 10 to 20 degrees [15]. Moderate IS is characterized by a Cobb angle of 25 to 40 degrees, which is indicated for non-operative treatment [16,17], and a Cobb angle from 21 to 35 degrees [15]. We consider as mild curves those with a Cobb angle above 10 but less than 20 degrees and as moderate those with a Cobb angle from 20 to 35–40 degrees. In these curves, especially in the mild ones, the rotation of the apical vertebrae is only a few degrees [18]. This morphology is very important for the measurements in the frontal and sagittal plane, which is minimally affected. This fact results in more reliable measurements, and it is very important for our study.

The degrees of Cobb angle in Table 1 and Table 2, presented by gender and curve type, for the 83 scoliotics of this study showed that the sample is suitable to enable us to draw conclusions on mild and moderate IS.

The segmental RI (a radiological measurement in the sagittal plane) and the RHD in terms of the measured ATR (a scoliometry/surface topography measurement) at all 83 cases with IS were significantly correlated to the segmental RI at only three levels, namely T6, T7 and T8, presented in Table 3. However, for the 25 scoliotics with a thoracic curve, the significance of the correlation was not only stronger, but it was also found at seven vertebral levels, namely T6, T7, T8, T9, T10, T11 and T12, presented in Table 4. For the thoracolumbar and lumbar curves, this correlation was not significant at any level. This implies the leading role of the RC, especially for the development of thoracic spinal deformity.

In mild and moderate IS curves, the vertebral rotation is minimal. Thus, rib cage deformity can generally be attributed to the asymmetric rib growth and to their deformation, and not to the vertebral rotation, as the rotation at this stage is minimal. Therefore, at any level from T1 to T12, a value of segmental RI equal or greater than 1.45–1.50 mainly reflects a significantly asymmetrical DRC, a fact indicating a remarkable asymmetrical growth of a pair of ribs at this spinal level. Therefore, this value of RI represents an increasing and progressive rib cage deformity.

The term “pattern of segmental RI asymmetry” is used to indicate the number of rib levels, from T1 to T12, with the above coined severe asymmetry, namely equal or more than 1.45–1.50.

The comparison of the three types of curve groups in female patients, using the Kruskal–Wallis test, showed significant differences between groups at T8, T9 and T12 vertebral levels. The post hoc analysis showed significant differences between thoracic and lumbar curves at T8, between thoracic and thoracolumbar curves at T9 and between thoracolumbar and lumbar curves at T12, as presented in Table 5.

In female patients with thoracic curves, presented in Table 5, the pattern of segmental RI asymmetry was present in eight levels from T3 to T10 (RI = 1.59–1.75); in thoracolumbar curves, the pattern of segmental RI asymmetry was present in four levels, from T2 to T5 (RI = 1.46–1.67), while in lumbar curves the RI did not exceed the value of 1.45 in all T1–T12 levels (Table 5).

In male patients with thoracic curves (Table 6), the pattern of segmental RI asymmetry was present in six levels, from T6 to T11 (RI = 1.51–1.75), i.e., at lower thoracic levels compared to the female pattern of RI asymmetry in thoracic curves. The pattern of segmental RI asymmetry with thoracolumbar curves was completely present in nine levels, from T3 to T5 (1.50–1.52) and T7 to T12 (1.58–1.70), i.e., in a much lower and more extended number of rib-pair levels compared to the RI pattern levels of asymmetry in female patients. In lumbar curves, the RI in male patients did not exceed the value of 1.44. These findings validate the role of rib cage deformity for the development of idiopathic scoliosis.

In male patients, the comparison of the RI in the three curve type groups, using the Kruskal–Wallis test, showed no significant difference between them (Table 6).

The lumbar curves in both female and male patients were minimally or not at all influenced by the RC deformity, but in contrast, the RC deformity had a great impact on the development of the thoracic curves in female patients, and unexpectedly on the thoracolumbar curves in male patients.

The further analysis of the segmental RI and Cobb angle correlations showed the following interesting findings.

The segmental RI by curve type and gender among all the three curve types—thoracic, thoracolumbar and lumbar—in female and male patients together, using the non-parametric Kruskal–Wallis test [**r**esults are presented as median (IQR)], showed no significant differences [the correlation was significant at the 0.05 level (2-tailed)] among the three groups.

The same analysis for scoliotic female patients only [using the Kruskal–Wallis test and the post hoc Mann–Whitney U-test] showed significant differences between groups at T8, T9 and T12 vertebral levels. The post hoc analysis showed significant differences between thoracic and thoracolumbar at T9 and at T12 between thoracolumbar and lumbar at T12 (Table 3).

The same analysis for scoliotic male patients only showed no significant differences between groups (Table 5).

These findings emphasize the importance of the role of RC asymmetry in relation to spinal deformity, mainly in the girls.

Published articles from previous decades sporadically reported that children could have surface deformity in terms of RHD without deformity in their spine [19,20]. Nissinen et al., 1989 [21,22], stated that “hump size was found to be the most powerful predictor of scoliosis. Large humps were more prevalent among non-scoliotic children that subsequently developed IS”. Additionally, Nissinen et al., 1993 [21,22], stated that asymmetric children with a hump deformity but without radiographically diagnosed scoliosis will develop IS during FU with an odds ratio of 1.72 in boys and 1.55 in girls.

The effect of growth was found to play a key role in the correlation between thoracic and spinal deformity in girls with IS [2]. Younger girls may be asymmetric in terms of RHD without having a spinal deformity (scoliosis). In older girls, a statistically significant correlation of thoracic and spinal deformity does exist, but this is not the case in younger girls. These findings implicate the important role of the RC in IS, because they show that RHD deformity precedes the spinal one in scoliogenesis of mild and moderate IS. This correlation was demonstrated by studying the RI at the most extended point of the most extending rib contour (convex) in the SLRs [4]. This finding inspired us to further investigate the role of RI at all rib-vertebral levels; that is, to assess the RI segmentally from T1 to T12 in RC, and to examine the relationship that these segmental RIs may have to the scoliogenesis of IS.

The findings of this study are in line with those reported by Prof. Sevastik’s research work, pertinent to scoliosis aetiology, emphasizing the important role of the rib cage in scoliosis development. Therefore, the benefit of this study is that its findings shed more light on the theory of asymmetric function of the ANS, reported by Prof. Sevastik and his team [23,24,25,26,27,28,29,30,31,32,33,34,35], and they also support a physiological approach to the surgical treatment of progressive early IS [36].

Investigating further the effect of growth on the relationship of RC with the spinal deformity in terms of the Cobb angle, we analyzed the segmentally measured RI from T1 to T12. For this purpose, the scoliotics were split into two age cohorts, namely (a) scoliotics less than 13 and more than 13 years of age, and (b) scoliotics less than 14 and more than 14 years of age, as presented in Table 7. It was thus feasible to recognize accurately the cut-off age (of the examined scoliotics) at which the segmental RI starts correlating or shows a stronger correlation to the spinal deformity, namely when the thoracic deformity starts to dictate the development of spinal deformity, in terms of Cobb angle.

Before any splitting of the ages, the significant correlation of Cobb angle with the segmental RI (Table 7) existed in thoracic curves T10, 11 and 12 (*p* = 0.462, 0.465, 0.547), respectively; in thoracolumbar curves, it correlated negatively at T4 only (*p* = 0.031), and in lumbar curves at T9 only (*p* = 0.046); Table 7.

After splitting the scoliotic boys and girls in two cohorts by age, namely over and less than 13 years, it was found that, as in Table 7:

In scoliotics over 13 years of age with thoracic curves, the Pearson correlation coefficient was significant at T11 and T12 (*p* = 0.045, 0.043).

In scoliotics less than 13 years of age with thoracic curves, the Pearson correlation coefficient was not significant.

In scoliotics less than 13 years of age with thoracolumbar curves, the correlation was significant at T4 (*p* = 0.026), and in those over 13 years of age at T6–9 (*p* = 0.039, 0.030, 0.014, 0.019), respectively.

In scoliotics less than 13 years of age with lumbar curves, the correlations at T1 and T2 were *p* = 0.010 and 0.001, and in those over 13 years of age at T8, 9 and T11 they were *p* = 0.036, 0.005, 0.045, respectively.

Similarly, by splitting the ages of scoliotics at over and less than 14 years of age, it was found that, as in Table 7:

In thoracic curves of patients over 14 years of age, the Pearson correlation coefficient was significant at T8–11 (*p* = 0.016, 0.022, 0.016, 0.009).

In thoracolumbar curves of patients less than 14 years of age at T4, the Pearson corr. coeff. was negatively correlated (*p* = 0.004), and in those over 14 years of age at T1 (*p* = 0.046), respectively.

Finally, in lumbar curves of patients less than 14 years of age, there was a correlation at T8 *p* = 0.020.

From this analysis, it is evident that the cut-off point of ages of the examined scoliotics was 14 years, when the segmental RI showed a stronger correlation to the spinal deformity, namely when the thoracic deformity correlated to the developing thoracic spinal deformity, in terms of Cobb angle. This is in accordance with what was earlier found [26].

The RC asymmetry described in this study of segmental RI is also in accordance with the Nottingham concept of IS pathogenesis. RC deformity is mainly due to asymmetric rib growth, and its deformation affects the rotation defending system of the thorax of the “flag pole dinner plate” concept of this theory, leading to scoliosis [37].

The results of the comparison of the segmental RI in the asymmetric but non-scoliotic children to the RI of scoliotics by curve (thoracic, thoracolumbar, lumbar) in boys and girls, interestingly, showed that there was no significant difference between groups (non-scoliotic to thoracic, non-scoliotic to thoracolumbar, non-scoliotic to lumbar). However, considering the fact that a good number of these asymmetric referred children will develop IS [21,22], our recommendation is not to discharge them from the scoliosis outpatient department but to follow them up for a longer period of time. This recommendation is also one of the benefits of this segmental rib index study.

A limitation of this report is the small number of curve types in the studied scoliotic children and adolescents. However, the results of the analysis are in line with those of a previous publication, see [2], where the included female sample was larger. Additionally, the results are based on two-dimensional radiography. Currently, three-dimensional analysis is more frequently used as a procedure to study the morphology of scoliotic curvatures [38,39,40,41,42], because any study based exclusively on the coronal or sagittal plane has its limitations. However, the most important and frequently used radiological parameters are designed and measured on A-P and lateral radiographs (i.e., Cobb, Mehta RVAs, Perdriolle angles).

The benefits of using the RI and segmental RI method are described above. RI can also be used as a surrogate for scoliometry [7]. RI is a radiographic measurement of asymmetry in the standing position, while the scoliometer measures clinically truncal asymmetry in flexion. It was found that a change from a flexed position to a standing position resulted in a reduction in trunk asymmetry [43]. If patients had RI asymmetry measured on their standing radiographs, the flexion scoliometry reading would have been greater. Therefore, segmental RI can safely be used as a strong substitute for scoliometric measurement and can estimate the severity of thoracic deformity [43].

One other benefit of using the RI and segmental RI is that the plain chest radiographs of children and adolescents, being easily available at medical archives, can effectively serve the segmental RI method, without the need for any other special radiographs and exposure to additional radiation. One additional benefit of the segmental RI method is its implementation not only in prospective but also in retrospective studies on non-operative and operative treatment of IS, using the existing initially obtained chest or spinal radiographs of IS patients, provided that the radiography is performed in a standard way.

The operation named costoplasty or thoracolpasty or pleuroplasty is the one that has been introduced to correct the deformity of the ribs in the chest, of the AIS patients [44,45,46,47]. In operable cases of IS with excessive hump, in addition to spinal surgery, costoplasty is sometimes performed. The results of costoplasty are not always satisfactory, as is mentioned in the literature, ref. [48], because in some patients there is a persistent or remaining rib deformity. As a result, the patients and families are not satisfied with the operation. The explanation of this phenomenon is provided by Erkula et al. 2003. As ribs slope obliquely downwards, it is difficult to predict which vertebral levels one is making measurements for. This anatomical quirk makes it difficult to recognize the exact levels of the maximum rib deformity in patients with a severe hump, and also the corresponding vertebral level. They recommend performing a scanogram or a 3D reconstruction of the spine and ribs, which will help to define the exact level of the rib deformity that corresponds to a certain vertebral level [48]. Their findings are very much in line with what triggered us to introduce the segment RI method. The maximum segmental RI value in the above-described pattern of increased RI values, see Table 5 and Table 6, could replace the scanogram or the 3D reconstruction and help with the recognition of the exact level or levels of the rib deformity/ies that correspond to the certain vertebral level which must be costoplasted. Thus, the patients will have less exposure to radiation, and this is considered one important benefit of using the new segmental RI method [43].

## 5. Conclusions

In conclusion, this is the first report presenting the segmental rib index according to the location of the curves in thoracic, thoracolumbar and lumbar IS. Additionally, the above study presented results from data collected retrospectively from scoliotics suffering mild and moderate IS. The reported correlations of the surface deformity, in terms of scoliometry, and radiological deformity, in terms of radiography, of the scoliotics in this report show the significant impact of the RC on the spine. The RC seems to play a protagonistic role in the scoliogeny of IS in mild and moderate thoracic and thoracolumbar IS.

## Figures and Tables

**Figure 1 healthcare-11-03004-f001:**
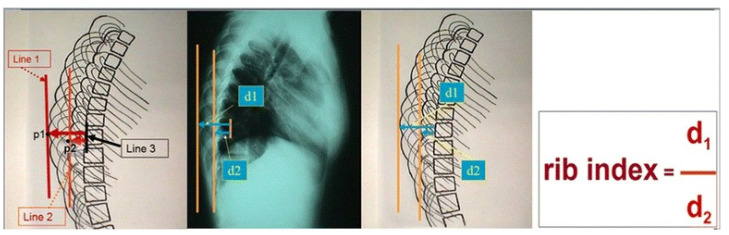
The way the RI is assessed on the standing lateral spinal radiographs (from Grivas 2014, [4]).

**Figure 2 healthcare-11-03004-f002:**
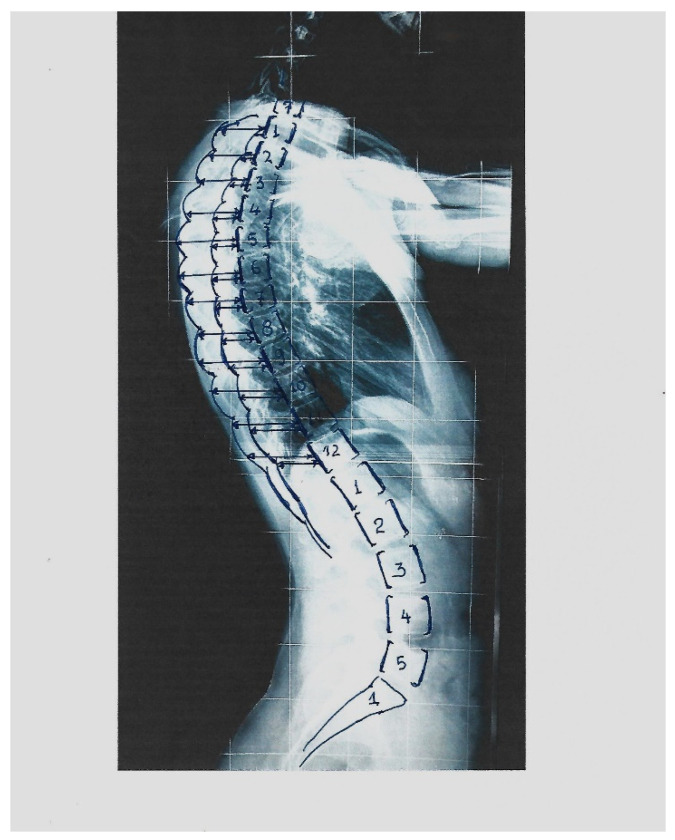
The way the segmental RI (SRI) is assessed on the standing lateral spinal radiographs.

**Table 1 healthcare-11-03004-t001:** Basic characteristics of patients.

	Total	Female	Male	*p* Value
N	83	63 (75.9%)	20 (24.1%)	
Age (years)	12.3 ± 2.7	12.5 ± 2.5	11.5 ± 3.5	0.241 ^a^
Risser stage	2 (4)	2 (4)	0 (2.8)	0.023 ^b^
Cobb angle	26.3 ± 10.8	27.7 ± 11.2	22.7 ± 7.8	0.068 ^a^
ATR	7.6 ± 4.3	8.0 ± 4.4	7.3 ± 4.6	0.514 ^a^

Results are shown as frequency (%), mean ± standard deviation, or median (IQR). ^a^ T test. ^b^ Mann–Whitney test.

**Table 2 healthcare-11-03004-t002:** Cobb angle by curve type and sex.

Gender	Curve Type	n	Mean ± SD
Male	Thoracic	6 (7.2%)	25.4 ± 8.1
	Thoracolumbar	6 (7.2%)	22.0 ± 10.4
	Lumbar	8 (9.6%)	21.1 ± 5.6
Female	Thoracic	19 (22.9%)	29.0 ± 12.2
	Thoracolumbar	26 (31.3%)	27.7 ± 9.8
	Lumbar	18 (21.7%)	26.2 ± 12.5

Results are shown as n (%) and mean ± standard deviation.

**Table 3 healthcare-11-03004-t003:** Segmental rib index and surface deformity (RHD) correlations in all the cases.

n = 83	Pearson Correlation	*p* Value
T6	0.292	0.007 *
T7	0.220	0.046 *
T8	0.240	0.029 *

* Correlation is statistically significant at the 0.05 level (2-tailed).

**Table 4 healthcare-11-03004-t004:** Segmental rib index and surface deformity (RHD) correlations in the 25 scoliosis patients with a thoracic curve.

n = 25	Pearson Correlation	*p* Value
T6	0.432	0.031 *
T7	0.419	0.037 *
T8	0.520	0.008 *
T9	0.424	0.034 *
T10	0.491	0.013 *
T11	0.556	0.004 *
T12	0.485	0.014 *

* Correlation is statistically significant at the 0.05 level (2-tailed).

**Table 5 healthcare-11-03004-t005:** The segmental rib index by curve type among all three curve types—thoracic, thoracolumbar and lumbar—in female patients.

Female	Thoracic n = 19	Thoracolumbar n = 26	Lumbar n = 18	*p* Value
T1	1.30 (0.48)	1.20 (0.43)	1.15 (0.50)	0.444 ^a^
T2	1.40 (0.54)	1.67 (0.72)	1.24 (0.71)	0.310 ^a^
T3	1.51 (0.51)	1.57 (0.59)	1.38 (0.64)	0.233 ^a^
T4	1.59 (0.58)	1.57 (0.40)	1.38 (0.51)	0.201 ^a^
T5	1.67 (0.95)	1.46 (0.35)	1.45 (0.48)	0.404 ^a^
T6	1.60 (0.60)	1.42 (0.40)	1.40 (0.42)	0.322 ^a^
T7	1.54 (0.51)	1.40 (0.29)	1.39 (0.47)	0.133 ^a^
T8	1.60 (0.50)	1.39 (0.30)	1.34 (0.45)	0.054 ^a^
T9	1.73 (0.52) ^TL^	1.38 (0.33)	1.42 (0.41)	0.018 ^a^
T10	1.75 (0.57)	1.42 (0.55)	1.20 (0.66)	0.052 ^a^
T11	1.43 (0.70)	1.49 (0.50)	1.20 (0.54)	0.091 ^a^
T12	1.22 (0.67)	1.41 (0.50) ^L^	1.11 (0.32)	0.021 ^a^
**Female**	**Thoracic vs. Thoracolumbar**	**Thoracic vs. Lumbar**	**Thoracolumbar vs. Lumbar**	
T9	0.011 *	-	-	
T12	-	-	0.005 *	

Results are presented as median (IQR). ^a^ Kruskal–Wallis test, post hoc Mann–Whitney U test. ^TL^—Significant difference between examined group and thoracolumbar group. ^L^—Significant difference between examined group and lumbar group. * *p* value is statistically significant with Bonferroni correction at 0.017.

**Table 6 healthcare-11-03004-t006:** The segmental rib index by curve type among all three curve types—thoracic, thoracolumbar and lumbar— in male patients.

Male	Thoracic (n = 6)	TL (n = 6)	Lumbar (n = 8)	*p* Value
T1	1.11 (0.20)	1.18 (0.39)	1.21 (0.49)	0.487 ^a^
T2	1.13 (0.51)	1.29 (0.42)	1.32 (0.63)	0.768 ^a^
T3	1.26 (0.58)	1.52 (1.17)	1.36 (0.47)	0.691 ^a^
T4	1.26 (0.66)	1.50 (1.13)	1.44 (0.51)	0.372 ^a^
T5	1.22 (0.88)	1.50 (1.02)	1.36 (0.62)	0.215 ^a^
T6	1.52 (0.59)	1.39 (0.88)	1.26 (0.58)	0.352 ^a^
T7	1.43 (0.35)	1.58 (0.82)	1.32 (0.60)	0.331 ^a^
T8	1.47 (0.28)	1.61 (1.01)	1.39 (0.66)	0.411 ^a^
T9	1.50 (0.17)	1.61 (0.97)	1.40 (0.59)	0.371 ^a^
T10	1.53 (0.41)	1.70 (0.95)	1.41 (0.34)	0.300 ^a^
T11	1.43 (0.66)	1.59 (0.54)	1.29 (0.33)	0.219 ^a^
T12	1.19 (0.38)	1.63 (0.90)	1.38 (0.43)	0.274 ^a^

Results are presented as median [interquartile range (IQR)]. ^a^ Kruskal–Wallis test.

**Table 7 healthcare-11-03004-t007:** Comparison of the segmental RI index to the Cobb angle at all ages for boys and girls.

	All Patients	Age < 13	Age > 13	Age < 14	Age > 14
	r (*p* value)	r (*p* value)	r (*p* value)	r (*p* value)	r (*p* value)
Thoracic curve	n = 25	n = 13	n = 12	n = 14	n = 11
T8	-	-	-	-	0.703 (0.016)
T9	-	-	-	-	0.679 (0.022)
T10	0.462 (0.020)	-	-	-	0.703 (0.016)
T11	0.465 (0.019)	-	0.587 (0.045)	-	0.745 (0.009)
T12	0.547 (0.005)	-	0.591 (0.043)	-	-
Thoracolumbar curve	n = 32	n = 15	n = 17	n = 19	n = 10
T1	-	-	-	-	0.642 (0.046)
T4	−0.388 (0.031)	−0.571 (0.026)	-	−0.623 (0.004)	-
T6	-	-	−0.503 (0.039)	-	-
T7	-	-	−0.527 (0.030)	-	-
T8	-	-	−0.582 (0.014)	-	-
T9	-	-	−0.561 (0.019)	-	-
Lumbar curve	n = 26	n = 10	n = 16	n = 10	n = 16
T1	-	−0.835 (0.010)	-	-	-
T2	-	−0.938 (0.001)	-	-	-
T8	-	-	0.527 (0.036)	0.613 (0.020)	-
T9	0.394 (0.046)	-	0.662 (0.005)	-	-
T11	-	-	0.507 (0.045)	-	-

Results are presented as Pearson correlation coefficient = r (*p* value).

## Data Availability

Data are available on demand.

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
