# Peer review of "Segmental Rib Index and Spinal Deformity: Scoliogenic Implications"

_healthcare, 2023, doi:10.3390/healthcare11223004_

Round 1

Reviewer 1 Report

Comments and Suggestions for Authors

Dear authors,

Thank you for your updated manuscript. I see that many of my comments have been addressed, which have improved the manuscript in my opinion.

However, I still have some major concerns regarding the scientific soundness of this paper.

The methods section regarding the statistical analysis and data sources and the actual results section to me is still insufficient. I lack a description of the type of data included in this study, whether it is cross-sectional data or longitudinal, etc. I see that the authors have added that a post-hoc analysis has been done to assess a significant p-value between sub-groups. However, what I'm still missing is a post-hoc analysis to assess for repeated data manipulation and therefore an adjustment of the p-value, such as a Dunn test.

As far as I understand this paper uses cross-sectional data, meaning that no causal inference can be drawn from this data. Therefore the current conclusion cannot be stated the way it is written right now. Also, I feel that the discussions section, does not answer the intial research question raised in the introduction.

Last but not least, I recommend the reviewers to not make assumptions on the gender of reviewers given that this is both unnecessary and inappropriate.

I wish the authors all the best with their revision of this paper.

Author Response

Reviewer 1

Thank you for your updated manuscript. I see that many of my comments have been addressed, which have improved the manuscript in my opinion.

Dear Reviewer, thank you for your comments, and for your time you spent to review our submission.

However, I still have some major concerns regarding the scientific soundness of this paper.

The methods section regarding the statistical analysis and data sources and the actual results section to me is still insufficient. I lack a description of the type of data included in this study, whether it is cross-sectional data or longitudinal, etc. I see that the authors have added that a post-hoc analysis has been done to assess a significant p-value between sub-groups. However, what I'm still missing is a post-hoc analysis to assess for repeated data manipulation and therefore an adjustment of the p-value, such as a Dunn test.

Dear Reviewer, thank you for your comment, and for your time you spent to review our submission. The type of data included in this study cross-sectional data.

As far as the description of the type of data included in this study and the statistical Analysis we addressed your questions, see the text, page 2, at the section material and methods.

As far as I understand this paper uses cross-sectional data, meaning that no causal inference can be drawn from this data. Therefore, the current conclusion cannot be stated the way it is written right now. Also, I feel that the discussions section, does not answer the initial research question raised in the introduction.

In our discussion we highlight the limitation of our study. A limitation of this report is the small number of the curve types of the studied scoliotic children and adolescents. Yet, the results of the analysis are in line with those of a previous publication, see [26], where the included female sample was larger. Additionally, the results are based on a two dimensional radiography.  Currently three-dimensional analysis is more frequently used as a procedure to study the morphology of scoliotic curvatures, [42,23,44,45,46], because any study based exclusively on coronal or sagittal plane has its limitations. However, the most important and frequently used radiological parameters are designed and measured on A-P and lateral radiographs (i.e. Cobb, Mehta RVAs, Perdriolle angles).

Additionally, in the introduction we revised the last phrase and we stated that “Any significant relationship of these parameters would be very important for the biomechanical relations of the RC (RHD) with the spine deformity and may thus provide possible information about scoliogeny. Also the last phrase was revised as “RC seems to plays a protagonistic role in the scoliogeny of IS in mild and moderate thoracic and thoracolumbar IS.

Last but not least, I recommend the reviewers to not make assumptions on the gender of reviewers given that this is both unnecessary and inappropriate. Dear Reviewer, apologies if there was left the impression that we made an assumption on the gender of reviewer. This impression is not representing our motivation; we respect very much both genders! It was incorrect to write in our answer to our first revision to the reviewer “We thank the reviewer for expressing his opinion” but ought to write “We thank the reviewer for expressing her/his opinion”. It was actually the “demon of the printing press” J.

Reviewer 2 Report

Comments and Suggestions for Authors

Congratulations to the authors on completing a large project. Please see my comments below.

The 6th Author, Elias and siliadis, seems like a mistake. Based on References 2, 14, 21, 26, and 49, it seems this author should be written as Elias S. Vasiliadis. Please advise if otherwise or correct on the manuscript.

WHOLE MANUSCRIPT

It seems as if there is font in black and some in light blue. Please make sure all content is in standard black font.

There is content throughout the manuscript that is very inconsistent (e.g. random italicized, bold, and underlined parts). Please undo italicized, bold, and underlining in the body (non-title) parts.

It seems that you used commas (",") throughout your manuscript for a decimal in numbers instead of a period ("."). For example, 12,3+2,7 as opposed to 12.3+2.7. This is done correctly in other places (ie page 4, line 122, you wrote 7.7 + 4.5 degrees). Please correct this anywhere in the manuscript.

1. INTRODUCTION

2. MATERIALS AND METHODS

How were the measurements made? By the authors? If not, did the authors supervise the examiners? How many examiners performed the measurements included in this study?

Was a reliability study performed to ensure the accuracy of measurements? If not, please refer to QAREL 11-point checklist for appropriate steps to take to create a high-quality reliability study with low bias (https://bmcmedresmethodol.biomedcentral.com/articles/10.1186/1471-2288-13-111/tables/1).

page 2, line 79

Based on the previous subheading "The examined subjects" it seems that "The measurements" should be italicized.

page 2, line 101

Statistical analysis should not be a part of "The measurements".

This should be its own subheading like The examined subjects and The measurements.

3. RESULTS

pages 3-4, Table 2

There are 2 lines under Female that do not seem to be appropriate.

page 4, lines 135-144

The subheading title "Segmental Rib Index and Cobb angle correlations" should be italicized based on formatting elsewhere in the manuscript.

The rest of the content in these lines seems to be very inconsistent (e.g. random italicized and underlined parts). Please undo italicized and underlining in these lines.

page 4, line 146, Table 5

Please remove the underlining under "at females" and remove the comma "," after at females.

page 5, line 151, Table 6

Please remove the underlining under "at males" and remove the comma "," after at males.

page 5, lines 154-169

Please undo bolding, italicized, and underlining in these lines.

page 5, line 169

You write, "see Table 7 i, ii, iii"

There is no Table 7 i, ii, iii. You only have Table 7 (no i, ii, or iii). Please remove the i, ii, and iii.

page 6, Table 8

This is formatting differently throughout the table (e.g. differences in subtitles having bold, underlining, or italics or not). Please match the consistent formatting of Table 7.

page 7, line 195

Remove 6.b. from the beginning of this paragraph.

page 7, line 199, Table 9

Please remove the comma "," after patients.

page 7, line 206, Table 9

"at less than 14 years-old" is not italicized here and it is italicized in the rest of the table subheadings. Please be consistent throughout ALL tables. Everything should be uniform throughout the manuscript.

4. DISCUSSION AND CONCLUSIONS

pages 8-10, lines 231-340

There is content throughout this section that is very inconsistent (e.g. random italicized, bold, and underlined parts). Please undo italicized, bold, and underlining.

page 9, lines 321, 323, and 334

You write, "Table 7 i, ii, iii" and "Table 8 i, ii, iii" and "Table 9 i, ii, iii"

There is no Table 7 i, ii, iii, Table 8 i, ii, iii, or Table 9 i, ii, iii. You only have Table 7, Table 8, and Table 9 (no i, ii, or iii). Please remove the i, ii, and iii.

page 9, lines 332 and 336

P=

The P needs to be lowercase.

page 10, line 368

"...Haber et al 2020, [47]."

Please remove Haber et al 2020 from "...Haber et al 2020, [47]." because you are providing a source citation. This should look like:

RI could be used as a surrogate for scoliometry, [47].

page 10, line 370

Remove space before 48 in brackets. You have [ 48] and it needs to be [48].

page 10, line 375

"...Grivas et al 2023, [49]."

Please remove Grivas et al 2023 from "...Grivas et al 2023, [49]." because you are providing a source citation. This should look like:

Therefore, segmental RI can safely be used as a strong substitute for scoliometric measurement and can estimate the severity of thoracic deformity, [49].

***Please fix this anywhere else in the manuscript. Thank you.

REFERENCES

Several references are not formatted appropriately with correct spacing and such. Too many errors to go over 1 by 1. Go to Instructions for Authors, review references formatting, and review every single reference for proper formatting. Pay attention to what needs to be included or not (e.g. PMID, DOI, etc.), spacing, and proper punctuation. Additionally, Reference 4 is not in English.

Comments on the Quality of English Language

See Comments and Suggestions for Authors

Author Response

Reviewer 2

Congratulations to the authors on completing a large project. Please see my comments below. Dear Reviewer, thank you for your comment, and for your time you spent to review our submission.

The 6th Author, Elias and siliadis, seems like a mistake. Based on References 2, 14, 21, 26, and 49, it seems this author should be written as Elias S. Vasiliadis. Please advise if otherwise or correct on the manuscript. Thank you. We corrected and included the capital letter S for the name of Elias Vasiliadis. Elias S. Vasiliadis is the same person to Elias Vasiliadis!

 WHOLE MANUSCRIPT

It seems as if there is font in black and some in light blue. Please make sure all content is in standard black font.

Thank you. We made all content in standard black font.

 There is content throughout the manuscript that is very inconsistent (e.g. random italicized, bold, and underlined parts). Please undo italicized, bold, and underlining in the body (non-title) parts.

Thank you. We undid italicized, bold, and underlining in the body in non-title parts of the text.

It seems that you used commas (",") throughout your manuscript for a decimal in numbers instead of a period ("."). For example, 12,3+2,7 as opposed to 12.3+2.7. This is done correctly in other places (ie page 4, line 122, you wrote 7.7 + 4.5 degrees). Please correct this anywhere in the manuscript.

Thank you. We did this correction as recommended.

  1. INTRODUCTION
  2. MATERIALS AND METHODS

How were the measurements made? By the authors? If not, did the authors supervise the examiners? How many examiners performed the measurements included in this study? Thank you. The measurements for the segmental rib index were done by DL, the scoliometric measurements were done by TBG and SP see Author Contributions section and the text in the method section.

Was a reliability study performed to ensure the accuracy of measurements? Thank you. We mention in the text that we previously reported the reliability study for the rib index method [6] and for the scoliometer readings [14].

If not, please refer to QAREL 11-point checklist for appropriate steps to take to create a high-quality reliability study with low bias (https://bmcmedresmethodol.biomedcentral.com/articles/10.1186/1471-2288-13-111/tables/1).

 page 2, line 79. Based on the previous subheading "The examined subjects" it seems that "The measurements" should be italicized.  page 2, line 101. Thank you. We italicized it.

Statistical analysis should not be a part of "The measurements". This should be its own subheading like The examined subjects and The measurements. Thank you. Statistical analysis was made as a subheading and also italicized.

  1. RESULTS

pages 3-4, Table 2. There are 2 lines under Female that do not seem to be appropriate.   Thank you. We corrected it.

page 4, lines 135-144. The subheading title "Segmental Rib Index and Cobb angle correlations" should be italicized based on formatting elsewhere in the manuscript. Thank you. According to your previous recommendation (Please undo italicized, bold, and underlining in the body (non-title) parts), this is not italicized as it is not a title, but it is a subheading.

The rest of the content in these lines seems to be very inconsistent (e.g. random italicized and underlined parts). Please undo italicized and underlining in these lines.  Thank you. We made the correction.

 page 4, line 146, Table 5. Please remove the underlining under "at females" and remove the comma "," after at females.  Thank you. We made the correction.

page 5, line 151, Table 6.  Please remove the underlining under "at males" and remove the comma "," after at males.  Thank you. We made the correction.

page 5, lines 154-169. Please undo bolding, italicized, and underlining in these lines.  Thank you. We made the correction.

 page 6, line 190-192 Tables 7, 8 and 9 were  merged into one (number 7), and adjusted the corresponding text (explanations and descriptions). The numbers in the table 7 now are the same as before in the table 7,8 and 9, but more noticeable and clearer. 

  1. DISCUSSION AND CONCLUSIONS

pages 8-10, lines 231-340. There is content throughout this section that is very inconsistent (e.g. random italicized, bold, and underlined parts). Please undo italicized, bold, and underlining. Thank you. We corrected it.

 page 9, lines 332 and 336. P=. The P needs to be lowercase. Thank you. We corrected it.  

page 10, line 368. "...Haber et al 2020, [47]." Please remove Haber et al 2020 from "...Haber et al 2020, [47]." because you are providing a source citation. This should look like: RI could be used as a surrogate for scoliometry, [47]. Thank you. Now it is RI could be used as a surrogate for scoliometry, [47].

 page 10, line 370. Remove space before 48 in brackets. You have [ 48] and it needs to be [48]. Thank you. Corrected.

 page 10, line 375. "...Grivas et al 2023, [49].". Please remove Grivas et al 2023 from "...Grivas et al 2023, [49]." because you are providing a source citation. This should look like: Therefore, segmental RI can safely be used as a strong substitute for scoliometric measurement and can estimate the severity of thoracic deformity, [49]. Thank you. Now it is “Therefore, segmental RI can safely be used as a strong substitute for scoliometric measurement and can estimate the severity of thoracic deformity, [49].”

***Please fix this anywhere else in the manuscript. Thank you.

REFERENCES

Several references are not formatted appropriately with correct spacing and such. Too many errors to go over 1 by 1. Go to Instructions for Authors, review references formatting, and review every single reference for proper formatting. Pay attention to what needs to be included or not (e.g. PMID, DOI, etc.), spacing, and proper punctuation. Additionally, Reference 4 is not in English.

Thank you. Reference 4 was translated into English, and the rest of literature revised as recommended.

Reviewer 3 Report

Comments and Suggestions for Authors

Thank you very much for allowing me to review the submitted work - Segmental rib index and spinal deformity: aetiological implications. The authors should be applauded for their efforts, but I do have some concerns noted below: 

Intro - good

Methods - needs work: inclusion/exclusion criteria needs expansion as well as general study design. Was this a single-center, single-surgeon cohort? Who performed the measurements? Was there any follow up data recorded? Ie. basics must be covered.

Results - Overall, the main issues here lie with trying to obtain granular analyses from a relatively small cohort. Baseline comparison (esp. by sex) should be included. As evidenced by Table 6, n=6 or 7 results in a very underpowered analysis and conclusions cannot reliably be drawn from this small of a cohort split. Recommend either addition of patients, or narrowing the goals of the study to better control for this. Additionally, if any follow up data was recorded, this would be vital to include to correlate with clinical outcomes. 

Conclusions - relatively unsupported in its current state given the low power due to subdivision of data. 

Minor notes - is there any "significance" of the blue or bolded text? Was this simply a formatting issue? 

Thank you again to the authors and editors for allowing me to review this important work. 

Comments on the Quality of English Language

Some English spelling/grammar issues noted. Recommend reviewing with a formal English editor. 

Eg. "The benefits of using the RI and segmental RI method are numerous and are described above." (page 10, line 366)". This is not necessarily wrong, but rather abrupt and such language/formatting could be improved". Furthermore, the benefits are described below this stand-alone statement. 

Author Response

Reviewer 3

Dear Reviewer, thank you for your kind words and the comment, and for your time you spent to review our submission.

Thank you very much for allowing me to review the submitted work - Segmental rib index and spinal deformity: aetiological implications. The authors should be applauded for their efforts, but I do have some concerns noted below: 

Intro – good. Thank you.

Methods - needs work: inclusion/exclusion criteria need expansion as well as general study design. Was this a single-center, single-surgeon cohort? Who performed the measurements? Was there any follow up data recorded? Ie. basics must be covered.

Thank you.

“inclusion/exclusion criteria” were improved. This study is a multicenter retrospective cross sectional statistical study. We mention this now in the Material and Methods section. The cohort was formulated with subjected from the three centers. The centers are mention in the affiliation section of the co-authors. Mild scoliosis children and adolescents and particularly asymmetric but not scoliotic children are groups quit difficult to collect and a collaborative team may fulfill this need. The study is not aiming to study the follow-up. The colleagues who performed the measurements are describe in the revised text and in the authors contribution section.

Results - Overall, the main issues here lie with trying to obtain granular analyses from a relatively small cohort. Baseline comparison (esp. by sex) should be included. As evidenced by Table 6, n=6 or 7 results in a very underpowered analysis and conclusions cannot reliably be drawn from this small of a cohort split. Recommend either addition of patients, or narrowing the goals of the study to better control for this. Additionally, if any follow up data was recorded, this would be vital to include correlate with clinical outcomes. 

Thank you. In our discussion we highlight the small number of cases, but as we noted mild scoliosis children and adolescents and particularly asymmetric but not scoliotic children are groups quit difficult to collect and a collaborative team may fulfill this need.

It would be great having follow up data to include and correlate with clinical outcomes, but this is not in the aim of this study. May we conduct it in a future project. Thank you for this suggestion, this idea, as well.

Conclusions - relatively unsupported in its current state given the low power due to subdivision of data. Thank you. In the abstract we write: In summary, the results of this study may provide aetiological implications for IS, as far as the role of thorax is concerned.  In the conclusion we also revised the last sentence and we write:   RC seems to plays a protagonistic role in the scoliogeny of IS in mild and moderate thoracic and thoracolumbar IS.

Minor notes - is there any "significance" of the blue or bolded text? Was this simply a formatting issue? Thank you. The blue, bolded and underlined text was edited into only black letters.

Thank you again to the authors and editors for allowing me to review this important work. 

Comments on the Quality of English Language

Some English spelling/grammar issues noted. Recommend reviewing with a formal English editor. Thank you. The English spelling/grammar issues were addressed.

Eg. "The benefits of using the RI and segmental RI method are numerous and are described above." (page 10, line 366)". This is not necessarily wrong, but rather abrupt and such language/formatting could be improved". Furthermore, the benefits are described below this stand-alone statement. Thank you.  We revised the phrase.

Round 2

Reviewer 1 Report

Comments and Suggestions for Authors

Dear authors,

Thank you for allowing me to review the revised version of your manuscript. I see that great improvements have been made in the results section, and some in the methods section. However, to me the discussion and conclusion section remain insufficiently sustained. The discussion section has not been adapted after adapting the results section, for instance which I consider strange. Unfortunately, just changing the intensity of the relationship between scoliogenity and RI in the conclusion section is not sufficient. Causality cannot be inferred from cross-sectional data and it is unscientific to insinuate you can.

Comments on the Quality of English Language

Please improve the revised section for language. Thank you

Author Response

Reviewer 1

Thank you for allowing me to review the revised version of your manuscript. I see that great improvements have been made in the results section, and some in the methods section. However, to me the discussion and conclusion section remain insufficiently sustained. The discussion section has not been adapted after adapting the results section, for instance which I consider strange. Unfortunately, just changing the intensity of the relationship between scoliogenity and RI in the conclusion section is not sufficient. Causality cannot be inferred from cross-sectional data and it is unscientific to insinuate you can.

Dear reviewer, thank you for your comments. In the text we added in the methods section: Study design: This is a retrospective statistical study on cross-sectional data collected in Greece, Bosnia and Herzegovina and Serbia.

Yes, causality cannot be inferred from cross-sectional data and it is unscientific to insinuate we can. I wish I could find the real cause of idiopathic scoliosis!

The word Scoliogeny, according to renown late professor of Nottingham University RG Burwell, embraces the aetiology, the pathogenesis and the pathobiomechanics.  Therefore, when I use the word scoliogeny / scoliogenetic ect, as the aetiology /aetiological of IS, I refer to the pathogenesis and the pathobiomechanics.

For this reason, I changed the word aetiology with this of scoliogeny in the title and the text of the submission as: “Segmental rib index and spinal deformity: scoliogenic implications.

I hope that this will resolve any confusion. Therefore, in the text we describe what scoliogeny means. See page 2.

This paper does not describe the causality but the pathogenesis and the pathobiomechanics aspects of scoliogenesis.

So far, no single gene has been identified as the sole responsible one for scoliogeny and there is no conclusive evidence whether AIS is a single disease or a collection of multiple diseases. Research suggests that AIS may be influenced by genetic factors, environmental factors, and spinal growth and development. Additionally, a significant body of research indicates the presence of diverse morphological features, clinical presentations, and prognoses among AIS patients. Complexity and heterogeneity are key characteristics of AIS aetiology and phenotype, suggesting that AIS can be considered as a relatively complex group of diseases. Thus, the opinion that the association of spinal deformity with the rib deformity is more kind of a spectrum may be considered as an existing option. However, the reported results of this study in this review implement that the spectrum is more likely inclined to disassociation of the scoliotic and rib deformity type.

This is a study of the novel method of segmental rib index, based on our rib index method, which was presented internationally at 2000. The correlations found analyzing our data permit to make assumption of the impact of rib cage deformity on the central axis that is the spine in mild and moderate idiopathic scoliosis. Additionally, nowhere in the text is written that rib cage asymmetry is the cause of idiopathic scoliosis, but that it plays a patho-biomechanical role in scoliogenesis. The statistical analysis of our data led us to formulate this submission.   See page 2.  We write “For idiopathic scoliosis (IS) the word aetiology strictly means the factor(s) causing the AIS, pathogenesis the mode of origin of the morbid process, and pathomechanism / pathobiomechanics the sequence of events in the evolution of its structural and functional changes that result from the pathological process [9]. Prof RG Burwell of Nottingham, suggested the word scoliogeny as the collective noun to include aetiology, pathogenesis and pathomechanism/pathobiomechanics [10]. However, when scoliogeny is used in this study, we will only referrer to the pathogenesis and the pathobiomechanics of this condition.”

We also improved the English language.

Thank you for your positive impact in the quality of our paper.

Best Regards

Theo Grivas

Reviewer 3 Report

Comments and Suggestions for Authors

Thank you to the authors for addressing my concerns. I have no additional comments at this time. 

Comments on the Quality of English Language

There remain very minor syntactical errors, but none major.   

Author Response

Dear Reviewer

Thank you

The English language was improved.

Thank you for your time to review our submission
